# Objective evaluation of visual fatigue in patients with intermittent exotropia

**Masakazu Hirota** [1,2], **Kozue Yada**[3], **Takeshi Morimoto**[1,4], **Takao Endo**[3], **Tomomitsu Miyoshi**[5], **Suguru Miyagawa**[1,6,7], **Yoko Hirohara**[1,6,7], **Tatsuo Yamaguchi**[7], **Makoto Saika**[7], **Takashi Fujikado**[1,6]*

1 Department of Applied Visual Science, Osaka University Graduate School of Medicine, Suita, Osaka, Japan, 2 Department of Orthoptics, Teikyo University Faculty of Medical Technology, Itabashi, Tokyo, Japan, 3 Department of Ophthalmology, Osaka University Graduate School of Medicine, Suita, Osaka, Japan, 4 Department of Advanced Visual Neuroscience, Osaka University Graduate School of Medicine, Suita, Osaka, Japan, 5 Department of Integrative Physiology, Osaka University Graduate School of Medicine, Suita, Osaka, Japan, 6 Special Research Promotion Group, Osaka University Graduate School of Frontier Biosciences, Suita, Osaka, Japan, 7 Topcon Corporation, Tokyo, Japan

* fujikado@ophthal.med.osaka-u.ac.jp

**Data Availability Statement:** All relevant data are within the manuscript and its Supporting Information files.

**Funding:** Topcon Corporation provided support in the form of salaries for Suguru Miyagawa, Yoko

## Abstract

### Purpose

The purpose of this study was to evaluate the degree of visual fatigue in patients with intermittent exotropia (IXT) using the binocular fusion maintenance (BFM) test.

### Methods

Fourteen patients with IXT (32.1 ± 16.4 years) and 15 age-matched healthy volunteers (31.2 ± 9.3 years) participated in the study. BFM was assessed by measuring the transmittance of liquid crystals placed in front of the subject's nondominant eye at the instance when binocular fusion was broken and vergence eye movement was induced. A questionnaire on subjective symptoms was administered to the subjects before and after the visual task. The visual task consisted of a reciprocal movement between 67 and 40 cm.

### Results

The change [post–pre] of BFM was significantly lower in the IXT group (−0.185 ± 0.187) than in the control group (−0.030 ± 0.070) ($P$ = 0.010). The change of total subjective eye symptom score was significantly greater in the IXT group (2.28 ± 1.43) than in the control group (0.93 ± 1.27) ($P$ = 0.018). The reduction in BFM rate with increasing total subjective eye symptom score was significantly greater in the IXT group (−0.106 ± 0.017) than in the control group (−0.030 ± 0.013) ($P$ = 0.006).

### Conclusion

The present findings objectively showed that patients with IXT are at a greater risk of visual fatigue in comparison with healthy individuals.

Hirohara, Tatsuo Yamaguchi, and Makoto Saika but did not have any role in study design, data collection and analysis, the decision to publish, or preparation of the manuscript. The specific roles of these authors are articulated in the "Authors" section. This does not alter our adherence to the policies of PlosOne on sharing data and materials. Takashi Fujikado received research funding from Topcon Corporation. This study is supported by grants from an Asian CORE Program, Japan Society for the Promotion of Science (JSPS), Advanced Nano Photonics Research and Education Center in Asia (TF); the JSPS Core-to-Core Program, A, Advanced Research Networks (TF); the Ministry of Education, Culture, Sports, Science and Technology Japan, Grant-in-Aid for Scientific Research B, J16H05487 (TF); Research Fellowship for Young Scientists, JSPS, 17J01295 (MH). The funders had no role in study design, data collection and analysis, decision to publish, or preparation of the manuscript.

**Competing interests:** Masakazu Hirota, Assistant Professor in Teikyo University Faculty of Medical Technology: Patent. Takeshi Morimoto, Associate Professor in Osaka University Graduate School of Medicine: None. Takao Endo, Medical Doctor in Osaka University Graduate School of Medicine: None. Tomomitsu Miyoshi, Assistant Teacher in Osaka University Graduate School of Medicine: None. Suguru Miyagawa, Employee of Topcon Corporation: Employee (Topcon), Patent. Yoko Hirohara, Employee of Topcon Corporation: Employee (Topcon). Tatsuo Yamaguchi, Employee of Topcon Corporation: Employee (Topcon), Patent. Makoto Saika, Employee of Topcon Corporation: Employee (Topcon), Patent. Takashi Fujikado, Professor in Osaka University Graduate School of Frontier Biosciences: Financial support (Topcon), Patent. This does not alter our adherence to the policies of PlosOne on sharing data and materials.

## Introduction

Intermittent exotropia (IXT) is the most common form of child-onset exotropia[1–3]. Although children with IXT are less commonly symptomatic, adults with IXT commonly complain of visual fatigue (eye strain or asthenopia), blurred vision, headache, and diplopia [4]. Previous questionnaire survey studies revealed that visual fatigue is one of the most prevalent symptoms in patients with IXT[4, 5].

Visual fatigue is related to vergence and accommodation parameters[6, 7]. Vergence is the simultaneous movement of the eyes to align both eyes to obtain or maintain binocular vision, and accommodation is the process of varying the refractive power of lens to produce a focused image on the retina at different distances from the object. Normal binocular vision comprises vergence and accommodation systems that act simultaneously[8]. Patients with IXT require excessive convergence and accommodation to maintain binocular vision[9, 10]. Previous studies suggest that the eyes of patients with IXT tend to tire easily compared with those of healthy individuals[4, 5, 9, 10].

In our previous work[11], we developed a binocular fusion maintenance (BFM) test for objective determination of visual fatigue. The BFM test has good reproducibility, and BFM scores significantly decrease after a visual task, showing a negative correlation with subjective eye symptom scores. This test can evaluate visual fatigue while maintaining vergence and accommodative stimuli constant[12–16]. BFM is expected to be lower in patients with IXT than in healthy individuals, if the eyes of patients with IXT tend to tire easily compared with those of healthy individuals.

The purpose of this study was to objectively evaluate the degree of visual fatigue in patients with IXT using the BFM test.

## Methods

### Subjects

Fourteen patients with IXT with a mean age of 32.1 ± 16.4 (mean ± standard deviation) years (range, 13–60 years) were recruited from the Department of Ophthalmology, Osaka University Hospital, Osaka, Japan. All patients were diagnosed by a single ophthalmologist who is a strabismus specialist (TF).

Fifteen group age-matched healthy volunteers with a mean age of 31.2 ± 9.3 years (range, 21–51 years) were recruited as controls via online recruitment. All patients and control healthy volunteers underwent ophthalmological examinations that included visual acuity at a distance of 5.0 m, angle of deviation using the alternate prism cover test both at proximity (33 cm) and at a distance (5.0 m), and stereo acuity (Titmus Stereo Tests; Stereo Optical Co., Chicago, IL, USA). Minus and plus signs in the angle of deviation indicate exodeviation and esodeviation in the alternate prism cover test. Stereo acuity was converted to the logarithm of arcsecond (log arcsec).

Written informed consent was obtained from all subjects after the nature and possible complications of the study were explained to them. This investigation adhered to the tenets of the World Medical Association Declaration of Helsinki. The experimental protocol and consent procedures were approved by the Institutional Review Board of Osaka University Medical School (approval no. 15294–4).

### Binocular fusion maintenance

BFM can be assessed by reducing the intensity of incident light on one eye, which is defined by the number of photons, because the perceptive size of the retinal image depends on the

intensity of incident light[17]. BFM was measured using a custom-made binocular open-view Shack–Hartmann wavefront aberrometer [(BWFA); Topcon Co., Ltd., Tokyo, Japan] with an 840 nm infrared light[18]. The BWFA was equipped with an eye tracking system that was used to monitor the pupil and corneal reflection with a 940 nm infrared light. This instrument measured and recorded binocular eye movements, wavefront aberrations, and pupil size simultaneously at a sampling rate of 30 Hz. The binocular fusion break can be judged objectively using the eye tracker with the BWFA because one eye deviates in the exo- or eso-direction after the binocular fusion break [19].

The liquid crystal shutters [X-FOS (G2)-CE 2 × 2; LC-Tec Displays AB, Borlänge, Sweden] were placed between the BWFA and the subjects' eyes to reduce the intensity of incident light. The transmittance of the liquid crystal shutter was changed linearly from 0.07% to 23.0%. This change was averaged in the wavelengths between 430 and 720 nm and confirmed with a spectroradiometer (SR-LEDW; Topcon).

During calibration, the subjects were asked to fixate on eight horizontal asterisk targets on a calibration plate placed 50 cm in front of their eyes. The positions of these targets in the horizontal plane were −8.0˚, −5.7˚, −3.4˚, −1.1˚, +1.1˚, +3.4˚, +5.7˚, and +8.0˚. Using a calibration curve, the distance between the center of the pupil and the corneal reflection was translated into the angle of ocular rotation. The measurement error at 50 cm was 0.3˚ to 0.5˚ (interquartile range). The binocular eye movements were used to calculate vergence.

The measurement procedure followed the process described in our previous study[11]. The spherical and cylindrical errors in all subjects were corrected between 0.00 D and −0.20 D using objective values obtained from the BWFA at 5.0 m. An examiner asked the subject about the sharpness of the starburst target (33.3 arc minutes) on the printed plate at 33 cm and added plus lenses to both eyes equally until the target could be seen clearly and confirmed that the subject's corrected visual acuity in each eye at 33 cm was equal to or greater than 0.0 logMAR.

The subjects continued to fixate the starburst target that was same as the one used to correcting the refractive errors at the subject's eye level and for wavefront aberrations of second orders (accommodative response), and pupil diameter were measured and recorded continuously for 50 s. The transmittance of the liquid crystal shutter the nondominant eye, which was determined by a hole-in-the-card test, was set at 23.0% for 2 s and was then reduced sequentially by 1.15% every second. Between 22 and 27 s, the transmittance was maintained at 0.07%, after which it was increased by 1.15% every second and was finally maintained at 23.0% between 47 and 50 s. The transmittance for the dominant eye was sustained at 23.0% throughout the 50 s period. The BFM test evaluated the intensity of incident light ratio with both eyes during binocular fusion break and was conducted three times before and three times after the visual task.

## Near point of convergence and fusional vergence range

IXT is characterized by prolonged NPC and low fusional convergence at close distances. Thus, all subjects performed the NPC and fusional vergence range tests before and after the visual task.

To measure the NPC, the subject was instructed to fixate on an accommodative target. An examiner then moved the target from a far to a near position until the subject perceived diplopia or one eye deviated from the fusional position. The distance from the bridge of the nose to the breakpoint was measured with a ruler and was determined as the NPC. If the measured value was ≤ 1 cm, it was recorded as 1 cm[20].

To measure the fusional vergence range, the subject fixated on a target placed at a distance of 5.0 m, with full-corrected spectacles. A prism bar was placed in front of the nondominant eye. The diopter of the prism was increased until the subject perceived diplopia or one eye

deviated from the fusional position. The dioptric value of the breakpoint was determined as the fusional vergence range.

## Visual task

During the visual task, all subjects fixated on the target (white asterisk of 2 cm on a black board), which was placed in front of the eyes. The target moved reciprocally (back and forth) from 67 cm to 40 cm [range of 1 meter angle (MA)] with speed 0.5 MA/s using electric motor (Movie 1). One trial was defined as the three reciprocating motions. The patients and healthy volunteers performed the visual task with correction of refractive errors for 5.0 m. All subjects underwent four trials. The binocular eye movements and accommodative responses were measured simultaneously during the visual task. The visual task was completed within 2 min.

## Subjective symptoms questionnaire

All subjects were asked to complete a subjective symptoms questionnaire at the beginning and end of the examination. The questionnaire was the same as that described in our previous studies (S1 Fig)[11, 21–23]). Questions 1–3 [1, How tired are your eyes?; 2, How clear is your vision?; 3, How do your eyes feel (pain and/or dry eye)?] were designed to assess subjective eye symptoms, whereas Questions 4–7 (4, How tired is your back?; 5, How tired is you neck?; 6, How severe is your headache?; 7, How sleepy do you feel?) were designed to assess physical and psychological discomfort. Each question was scored from 0 to 4, and all subjects were asked to choose one score for each question. The subjects gave their responses after hearing the questions about overall fatigue, and not just eyes, to avoid bias. The subjective eye symptom scores (Q1, Q2, Q3) were summed up to obtain the subjective eye symptom score. The physical and psychological discomfort scores (Q4, Q5, Q6, Q7) were summed to obtain the total subjective physical and psychological discomfort score.

## Data analysis

Patients with IXT showed a wide variety of clinical features. Burian's classification of intermittent exotropia was based on the difference between distant exodeviation and near exodeviation and was categorized into the following three types, basic type (difference between distant exodeviation and near deviation <10 prism diopter (PD)), divergence excess type (distant exodeviation is > 10 PD greater than near exodeviation), and convergence insufficiency (CI) type (distant exodeviation is > 10 PD lower than near exodeviation) [24, 25]. Patients with IXT were classified into subgroups as per this classification.

Data on eye positions, aberrations, and pupil sizes in both eyes were exported to an Excel file. Data were excluded if the pupil diameter changed by more than 2 mm per frame due to blinking[26]. Data were also excluded if the pupil diameter changed by more than 0.2 mm per frame over an average of 11 points and a median of 5 points due to noise. The missing values were replaced by a linearly interpolated value.

The eye position data collected during the 50 s measurement periods were averaged over the three trials before and after the visual task. The binocular fusion break time ($T_B$) was calculated automatically from the nondominant eye movements based on the results of our previous study[11] using Python 3.6.5. $Base_{min}$ was determined as the average eye position over 2 seconds after beginning the measurement in which the transmittance of the liquid crystal shutter remained equal between the right and left eye. $Base_{max}$ was determined as the average eye position in the nondominant eye over 2 seconds between 25 and 27 seconds in which the difference in the transmittance between the right and left eye was the largest. The amplitude in the deviation of the nondominant eye ($D_n$) was calculated as [$Base_{max}$ − $Base_{min}$]. Then, the points

at which the amplitude of deviation in the nondominant eye reached 10% and 90% of the total amplitude, designated as $0.1D_n$ and $0.9D_n$, respectively, were determined during the fusion break phase. A linear regression line was created using the nondominant eye position at $0.1 D_n$ and $0.9 D_n$. Then, $T_B$ was determined as the intersection between $Base_{min}$ and the linear regression line of the fusion break phase. BFM was calculated by the following equation:

$$\text{Binocular fusion maintenance (BFM)} = 1 - \frac{\text{Transmittance}_{\text{(nondominant eye)}} \text{ at } T_B}{\text{Transmittance}_{\text{(dominant eye)}}}$$

### Statistical analysis

Differences in BFM, NPC, fusional vergence range, and subjective symptom scores before and after the visual task were assessed by the Wilcoxon signed-rank test after assessment of normality by the Shapiro–Wilk test within the IXT and control groups.

To assess the significance of the differences between the IXT and control groups in the changes (post − pre) in BFM, NPC, fusional vergence range, total subjective scores of eye symptom (Q1 + Q2 + Q3), and physical and psychological discomfort (Q4 + Q5 + Q6 + Q7) were determined by the Mann–Whitney $U$ test after assessment of normality by the Shapiro–Wilk test. The same analysis was done conducted in subgroups of IXT.

The degree of visual fatigue was then evaluated. The relationships between changes in BFM and total subjective scores of eye symptom and physical and psychological discomfort in the IXT and control groups were analyzed, and the significance of differences in slope and intercept between the two groups was determined by the generalized linear mixed-effect model. The same analysis was done conducted in subgroups of IXT.

IBM SPSS Statistics v26 (IBM Corp., Armonk, NY, USA) was used to determine the significance of the differences, and a $P$ value $< 0.05$ was considered to indicate statistical significance.

### Results

In the IXT group (Table 1), 9 patients were basic type IXT and 5 patients were convergence insufficiency (CI) type IXT. Of the 14 patients with IXT, 3 underwent strabismus surgery at least 4 months before the study. The mean refractive error [spherical equivalent (SE)] of the right eye was −2.50 ± 3.13 (mean ± standard deviation) diopters (D) and that of the left eye was −2.46 ± 3.23 D. Best corrected visual acuity (BCVA) was equal to or greater than 0.0 logMAR (minimum angle of resolution) in all patients. The average angle of deviation was −20.0 ± 9.7 PD at proximity and −14.6 ± 13.4 PD at a distance. The mean stereo acuity was 1.84 ± 0.24 log arcsec.

In the control group (Table 2), the mean SE of the right eye was −2.60 ± 1.63 D and that of the left eye was −2.65 ± 1.76 D. BCVA was equal to or greater than 0.0 logMAR in all subjects. The average angle of deviation was −4.8 ± 5.3 PD at proximity and −0.6 ± 5.2 PD at a distance. All healthy individuals had a stereo acuity of 1.60 log arcsec.

Representative vergence and accommodative response data (healthy volunteer 14 and patient 10) during the performance of the visual task are shown in Figs 1 and 2. The binocular coordination was checked based on numerical data and anterior video data. Healthy volunteer maintained binocular coordination during the visual task (Fig 1). However, binocular coordination in the patient with IXT was gradually disrupted during the visual task (Fig 2a). The target followability of the nondominant eye deteriorated in all patients. The accommodative response also did not follow the accommodative stimulus after binocular coordination was

**Table 1. Demographics of the intermittent exotropia (IXT) group.**

| Patient | Classification | Age (y) | SE (D) | | Angle of deviation (PD) | | Stereo acuity (log arcsec) |
|---|---|---|---|---|---|---|---|
| | | | RE | LE | Near | Far | |
| P1 | CI type | 60 | 0.25 | 0.00 | −18.0 | −8.0 | 2.00 |
| P2 | Basic type | 13 | −6.50 | −7.25 | −16.0 | −10.0 | 2.15 |
| P3 | Basic type | 52 | 4.50 | 4.25 | −6.0 | −4.0 | 2.00 |
| P4 | Basic type | 25 | −4.25 | −3.50 | −12.0 | −4.0 | 1.70 |
| P5 | Basic type | 25 | −3.00 | −1.75 | −14.0 | −12.0 | 1.60 |
| P6 | CI type | 21 | −0.25 | −0.25 | −14.0 | −4.0 | 1.60 |
| P7 | CI type | 34 | −5.50 | −4.50 | −18.0 | −6.0 | 1.60 |
| P8 | CI type | 30 | −5.25 | −5.25 | −25.0 | −14.0 | 1.60 |
| P9 | Basic type | 23 | −3.50 | −4.50 | −25.0 | −30.0 | 1.70 |
| P10 | CI type | 30 | −2.75 | −3.25 | −30.0 | −8.0 | 1.70 |
| P11 | Basic type | 13 | −6.25 | −7.25 | −45.0 | −50.0 | 2.15 |
| P12 | Basic type | 15 | −0.50 | 0.00 | −20.0 | −25.0 | 1.60 |
| P13 | Basic type | 55 | 0.75 | 0.50 | −25.0 | −25.0 | 2.15 |
| P14 | Basic type | 53 | −2.75 | −1.75 | −12.0 | −4.0 | 2.15 |

The error term is the standard deviation. Minus and plus signs in the angle of deviation indicate exodeviation and esodeviation, respectively. CI, convergence insufficiency; y, years; SE, spherical equivalent; D, diopter; PD, prism diopter; RE, right eye; LE, left eye; log arcsec, logarithm of arcsecond.

disrupted in the patient with IXT (Fig 2b). All healthy volunteers maintained binocular coordination during the visual task. However, binocular coordination was gradually disrupted in all patients with IXT during the visual task.

In the IXT group, BFM was significantly lower for the postvisual task than for the previsual task (0.729 ± 0.252 vs. 0.915 ± 0.119; $P = 0.003$; Fig 3a, Table 3). NPC was significantly reduced

**Table 2. Demographics of the control group.**

| Healthy volunteer | Age (y) | SE (D) | | Angle of deviation (PD) | | Stereo acuity (log arcsec) |
|---|---|---|---|---|---|---|
| | | RE | LE | Near | Far | |
| H1 | 35 | −0.75 | −0.50 | −6.0 | 0.0 | 1.60 |
| H2 | 21 | −3.00 | −3.00 | −2.0 | 2.0 | 1.60 |
| H3 | 25 | −2.25 | −2.25 | −2.0 | −2.0 | 1.60 |
| H4 | 23 | 0.00 | 0.00 | −2.0 | 0.0 | 1.60 |
| H5 | 24 | −3.25 | −3.25 | −12.0 | −1.0 | 1.60 |
| H6 | 44 | −2.50 | −2.50 | −2.0 | 0.0 | 1.60 |
| H7 | 44 | −1.75 | −3.00 | −4.0 | 0.0 | 1.60 |
| H8 | 51 | −4.25 | −5.00 | 0.0 | 0.0 | 1.60 |
| H9 | 25 | −3.75 | −4.00 | 2.0 | 3.0 | 1.60 |
| H10 | 30 | −4.50 | −4.50 | −10.0 | 0.0 | 1.60 |
| H11 | 24 | −3.00 | −1.75 | −4.0 | 0.0 | 1.60 |
| H12 | 27 | 0.00 | 0.00 | −12.0 | −4.0 | 1.60 |
| H13 | 40 | −2.75 | −2.50 | 4.0 | 2.0 | 1.60 |
| H14 | 30 | −1.50 | −1.50 | −10.0 | −6.0 | 1.60 |
| H15 | 25 | −5.75 | −6.00 | −12.0 | −4.0 | 1.60 |

The error term is the standard deviation. Minus and plus signs in the angle of deviation indicate exodeviation and esodeviation, respectively.

y, years; SE, spherical equivalent; D, diopter; PD, prism diopter; RE, right eye; LE, left eye; NPC, near point convergence; log arcsec, logarithm of arcsecond.

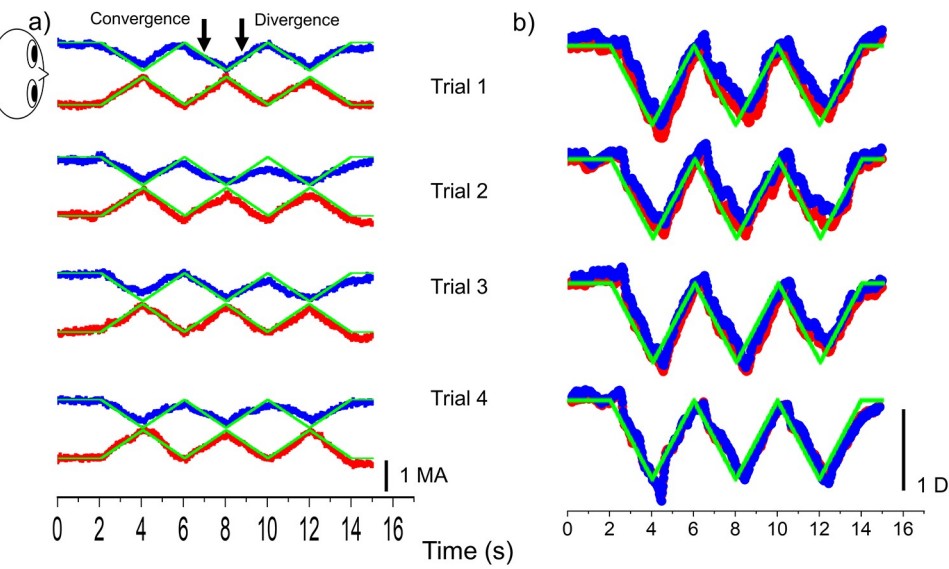

**Fig 1. Vergence (a) and accommodative (b) responses of a healthy volunteer stimulus in the visual task.** The blue, red, and green squares indicate the left eye, right eye, and stimuli, respectively. The healthy volunteer maintained binocular coordination during the visual task. MA, meter angle; D, diopter.

for the postvisual task than for the previsual task (4.7 ± 3.2 cm vs. 5.6 ± 3.8 cm; $P = 0.012$; Fig 3b, Table 3, and S1 and S2 Tables). Fusional vergence range was not significantly different between the pre- and postvisual task (23.6 ± 8.7 PD vs. 21.1 ± 10.8 PD; $P = 0.134$; Fig 3c, Table 3, and S1 and S2 Tables). Subjective eye symptom scores (Q1, Q2, and Q3) were significantly worse in the postvisual task than in the previsual task (Q1, $P = 0.003$; Q2, $P = 0.034$; Q3, $P = 0.002$; Table 3, and S1 and S2 Tables). The quartiles of BFM, NPC, and fusional vergence

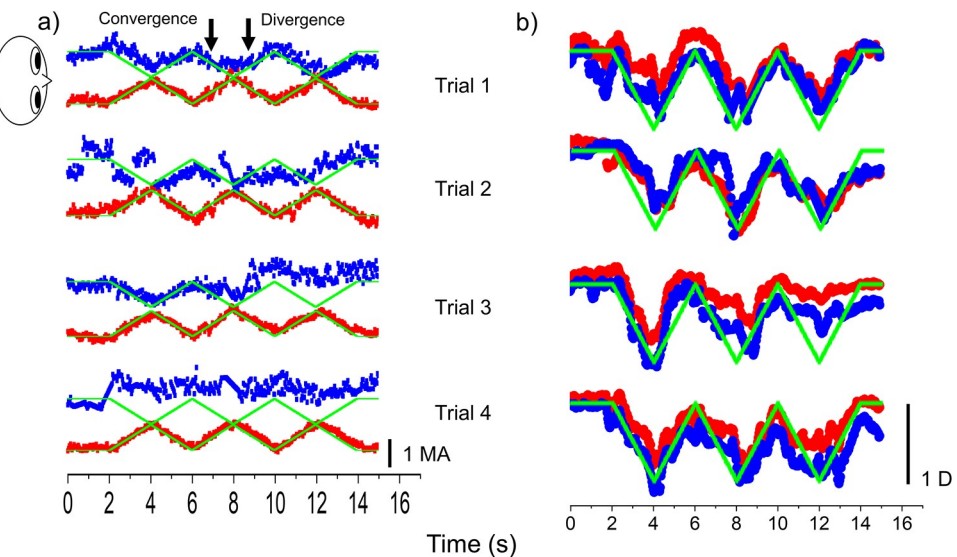

**Fig 2. Vergence (a) and accommodative (b) responses of a patient with intermittent exotropia (IXT) stimuli in the visual task.** The blue, red, and green squares indicate the left eye, right eye, and stimuli, respectively. Binocular coordination in the patient with IXT was gradually disrupted during the visual task. MA, meter angle; D, diopter.

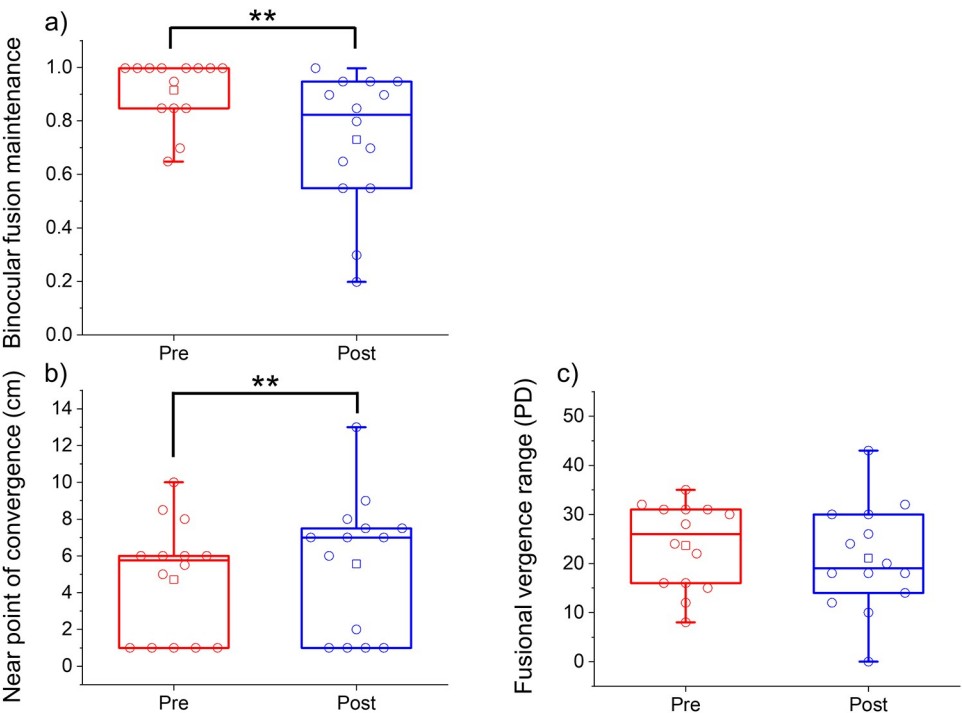

**Fig 3. Binocular fusion maintenance (BFM) (a), near point of convergence (NPC) (b), and fusional vergence range (c) before (red) and after (blue) the visual task within the intermittent exotropia (IXT) group.** The red and blue circles indicate individual BFM, NPC, and fusional vergence range values within the IXT group. The red and blue squares indicate the mean values for all the patients. $^{**}P < 0.01$, Wilcoxon signed-rank test.

range in the previsual and postvisual task were 0.850 and 0.950 vs. 0.555 and 0.950, 1.0 and 6.5 vs. 1.0 and 7.5, and 16.0 and 32.0 vs. 14.0 and 30.0, respectively. Furthermore, the medians of BFM, NPC, and fusional vergence range were 1.00 vs. 0.822, 5.8 vs. 7.0, and 26.0 vs. 19.0, respectively, and 95% confidence intervals of BFM, NPC, and fusional vergence range were 0.846–0.984 vs. 0.583–0.876, 2.9–6.5 vs. 3.4–7.7, and 18.6–28.7 vs. 14.8–27.3, respectively.

**Table 3. Mean results for the intermittent exotropia (IXT) group.**

| Test | Previsual task | Postvisual task | *P* value |
|---|---|---|---|
| BFM | 0.915 ± 0.119 | 0.729 ± 0.252 | * 0.003 |
| NPC (cm) | 4.7 ± 3.2 | 5.6 ± 3.8 | * 0.012 |
| Fusional vergence range (PD) | 23.6 ± 8.7 | 21.1 ± 10.8 | 0.134 |
| Subjective symptom questionnaire | | | |
| Q1 | 2.07 ± 0.83 | 2.93 ± 0.83 | * 0.003 |
| Q2 | 1.14 ± 0.53 | 1.64 ± 1.01 | * 0.034 |
| Q3 | 2.00 ± 0.88 | 2.92 ± 0.92 | * 0.002 |
| Q4 | 2.28 ± 1.14 | 2.28 ± 1.07 | 0.99 |
| Q5 | 2.21 ± 1.31 | 2.21 ± 1.12 | 0.99 |
| Q6 | 1.21 ± 1.12 | 1.00 ± 0.96 | 0.32 |
| Q7 | 1.43 ± 1.01 | 1.78 ± 1.25 | 0.21 |

The error term is the standard deviation. The pre- and postvisual task differences were analyzed by the Wilcoxon signed-rank test. BFM, binocular fusion maintenance; NPC, near point of convergence; PD, prism diopter.

In the control group, BFM did not significantly change from before (0.947 ± 0.068) to after (0.917 ± 0.082; P = 0.139) the visual task (Fig 4a, Table 4, and S3 and S4 Tables). NPC was significantly lower for the postvisual task than for the previsual task (2.6 ± 2.1 cm vs. 1.8 ± 1.8 cm; P = 0.043; Fig 4b, Table 4, and S3 and S4 Tables). Fusional vergence range was not significantly different between the pre- (32.5 ± 8.9 PD) and postvisual task (33.4 ± 7.6 PD; P = 0.54; Fig 4c, Table 4, and S4 Tables). The subjective eye symptom score (Q1) was significantly worse in the postvisual task than in the previsual task (Q1, P = 0.013; Table 4, S3 and S4 Tables). The subjective eye symptom scores of Q2 (pre, 0.87 ± 0.64; post, 0.80 ± 0.41; P = 0.57; Table 4) and Q3 (pre, 1.13 ± 0.74; post, 1.46 ± 0.64; P = 0.096; Table 4) were not significantly different between the previsual and postvisual task. The quartiles of BFM, NPC, and fusional vergence range in the previsual and postvisual task were 0.900 and 1.00 vs. 0.850 and 1.00, 1.0 and 1.0 vs. 1.0 and 4.0, and 26.0 and 44.0 vs. 28.0 and 40.0, respectively. The medians of BFM, NPC, and fusional vergence range were 1.00 vs. 0.950, 1.0 vs. 1.0, and 30.0 vs. 33.0, respectively. Furthermore, 95% confidence intervals of BFM, NPC, and fusional vergence range were 0.909–0.984 vs. 0.872–0.962, 0.8–2.7 vs. 1.4–3.7, and 27.6–37.5 vs. 29.2–37.7, respectively.

## IXT vs. control

The change in BFM was significantly and negatively greater in the IXT group than in the control group (−0.185 ± 0.187 vs. −0.030 ± 0.070; P = 0.010; Fig 5a). The change in NPC (0.8 ± 1.0 cm vs. 0.8 ± 1.5 cm; P = 0.53; Fig 5b) and fusional vergence range (−2.6 ± 6.8 PD vs. 0.9 ± 5.1 PD; P = 0.077; Fig 5c) were not significantly different between the IXT and control group. The change in total subjective eye symptom score was significantly greater in the IXT group than

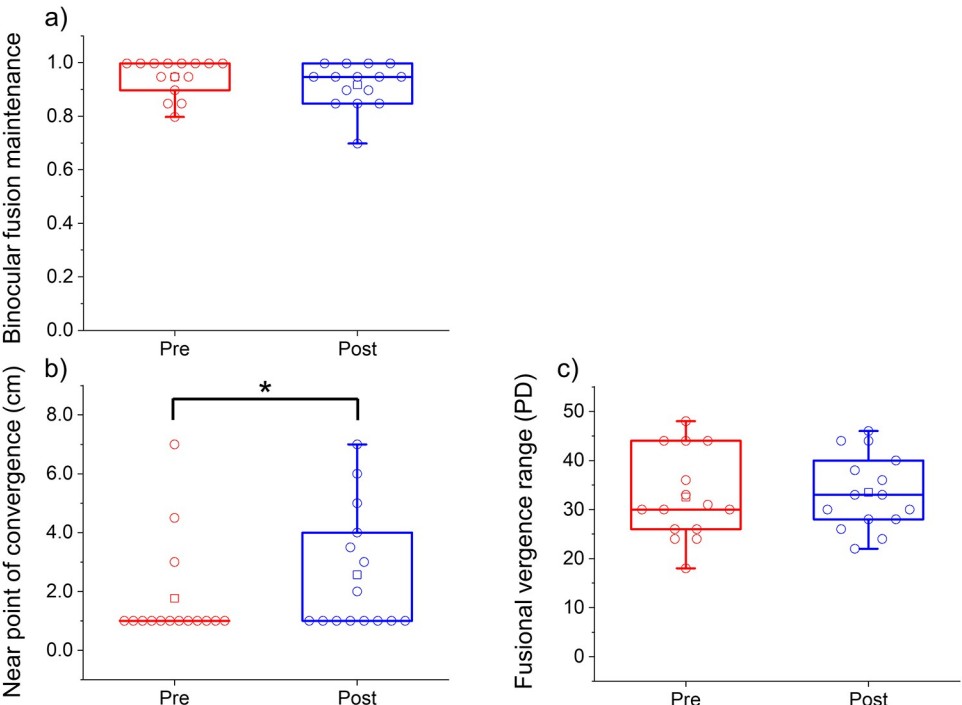

**Fig 4. Binocular fusion maintenance (BFM) (a), near point of convergence (NPC) (b), and fusional vergence range (c) before (red) and after (blue) the visual task within the control group.** The red and blue circles indicate individual BFM, NPC, and fusional vergence range values within the control group. The red and blue squares indicate the mean values for all the healthy volunteers. * P < 0.05, Wilcoxon signed-rank test.

**Table 4. Mean results for the control group.**

| Test | Previsual task | Postvisual task | P value |
|---|---|---|---|
| BFM | 0.947 ± 0.068 | 0.917 ± 0.082 | 0.139 |
| NPC (cm) | 1.8 ± 1.8 | 2.6 ± 2.1 | * 0.043 |
| Fusional vergence range (PD) | 32.5 ± 8.9 | 33.4 ± 7.6 | 0.54 |
| Subjective symptom questionnaire | | | |
| Q1 | 1.00 ± 0.76 | 1.67 ± 0.72 | * 0.013 |
| Q2 | 0.87 ± 0.64 | 0.80 ± 0.41 | 0.57 |
| Q3 | 1.13 ± 0.74 | 1.46 ± 0.64 | 0.096 |
| Q4 | 1.07 ± 0.79 | 1.20 ± 0.86 | 0.157 |
| Q5 | 1.00 ± 0.76 | 1.07 ± 0.79 | 0.57 |
| Q6 | 0.67 ± 0.62 | 0.67 ± 0.48 | 0.99 |
| Q7 | 1.00 ± 0.65 | 1.28 ± 1.08 | 0.21 |

The error term is the standard deviation. The pre- and postvisual task differences were analyzed by the Wilcoxon signed-rank test. BFM, binocular fusion maintenance; NPC, near point of convergence; PD, prism diopter.

in the control group (2.28 ± 1.43 vs. 0.93 ± 1.27; $P$ = 0.018; Fig 5d). The changes in total physical and psychological discomfort (0.11 ± 0.38 vs. 0.04 ± 0.38) were not significantly different between the two groups ($P$ = 0.56).

The quartiles of BFM, NPC, fusional vergence range, and total subjective eye symptom score were −0.038 and −0.300 vs. 0.000 and −0.100, 0.0 and 2.0 vs. 0.0 and 1.0, 0.5 and −5.8 vs. −2.0 and 3.0 and 1.0 and 3.3 vs. 0.0 and 2.0. The medians of BFM, NPC, fusional vergence

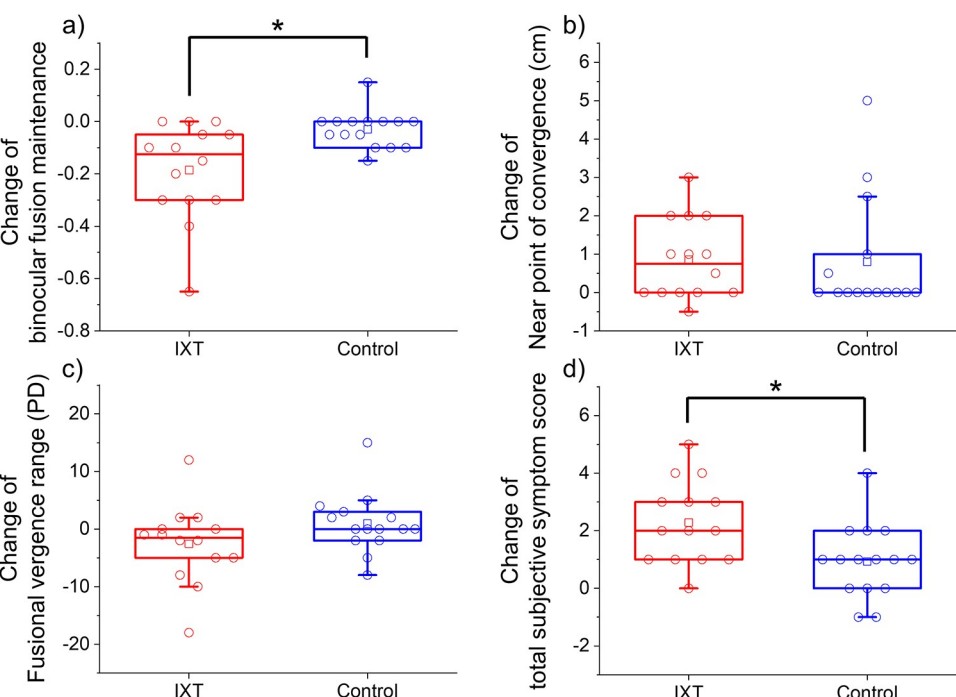

**Fig 5. Changes in binocular fusion maintenance (BFM) (a), near point of convergence (NPC) (b), fusional vergence range (c), and total subjective eye symptom score (b) in the intermittent exotropia (IXT) (red) and control (blue) groups.** The red and blue circles indicate individual BFM, NPC, fusional vergence range, and total subjective eye symptom scores (Q1 + Q2 + Q3) in the IXT and control groups. *$P$ < 0.05, Mann–Whitney $U$ test.

range, and total subjective eye symptom score were 1.000 vs. 0.000, 1.0 vs.0.0, 1.0 vs.0.0, and 30.0 vs. 1.0. The 95% confidence intervals of BFM, NPC, fusional vergence range, and total subjective eye symptom score were (−0.293 to −0.077 vs. −0.069 to 0.009), (0.3–1.5 vs. 0.0 to 1.6), (−6.5 to 1.4 vs. −1.9 to 3.8) and (1.4 to 3.1 vs. 0.2 to 1.6).

The change in BFM was significantly and negatively correlated with the change in total subjective eye symptom score in the IXT group ($R^2$ = 0.665, $P$ < 0.001) and the control group ($R^2$ = 0.292, $P$ = 0.038; Fig 6a). The slope was significantly and negatively steeper in the IXT group than in the control group (−0.106 ± 0.017 vs. −0.030 ± 0.019; $P$ = 0.006), but the intercept did not significantly differ between the groups ($P$ = 0.29; Fig 6a).

The change in NPC and fusional vergence range were not significantly correlated with the change in total subjective eye symptom score in the IXT group (NPC, $R^2$ = 0.017, $P$ = 0.66; fusional vergence range, $R^2$ = 0.069, $P$ = 0.37) and control group (NPC, $R^2$ = 0.002, $P$ = 0.87; fusional vergence range, $R^2$ = 0.000, $P$ = 0.99). The slopes in NPC (0.133 ± 0.260 vs. 0.057 ± 0.327; $P$ = 0.85) and fusional vergence range (−1.479 ± 1.168 vs. −0.003 ± 1.109; $P$ = 0.40) were not significantly different in the IXT and control group.

The changes in the BFM, NPC, and fusional vergence range were not significantly correlated with that in the total physical and psychological discomfort scores in the IXT group (BFM, $R^2$ = 0.206, $P$ = 0.102; NPC, $R^2$ = 0.004, $P$ = 0.71; fusional vergence range, $R^2$ = 0.002, $P$ = 0.83) and control group (BFM, $R^2$ = 0.193, $P$ = 0.101; NPC, $R^2$ = 0.011, $P$ = 0.72; fusional vergence range, $R^2$ = 0.001, $P$ = 0.98). The slopes in the BFM (−0.218 ± 0.123 vs. −0.077 ± 0.044; $P$ = 0.28), NPC (−0.279 ± 0.769 vs. −0.179 ± 1.047; $P$ = 0.94) and fusional vergence range (−0.108 ± 5.053 vs. −2.075 ± 3.508; $P$ = 0.75) were not significantly different for the IXT and control groups.

## Subgroup in IXT: Basic type vs. CI type

The changes in BFM, NPC, fusional vergence range, and total subjective symptom score were not significantly different between the patients with basic type IXT and the patients with CI type IXT (Table 4). The slope in BFM on total subjective eye symptom score was significantly and negatively steeper in the basic type IXT than in the CI type IXT (−0.131 ± 0.022 vs.

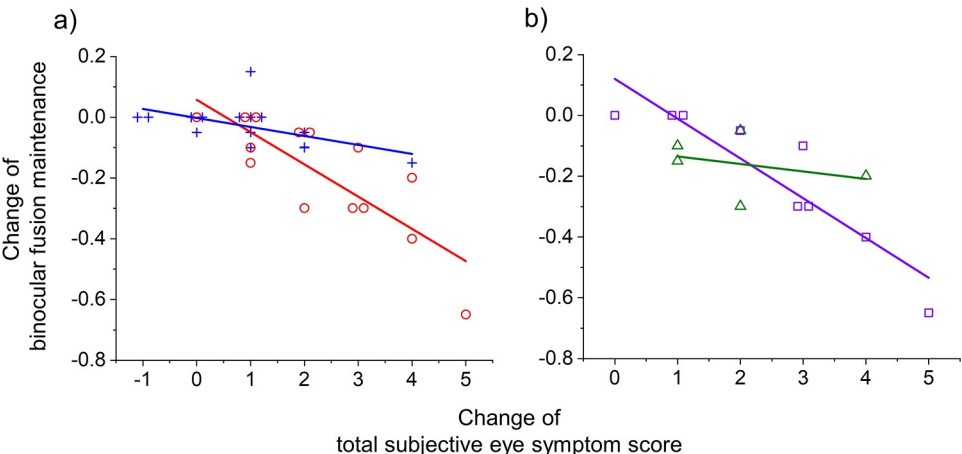

**Fig 6. Relationship between binocular fusion maintenance (BFM) and total subjective eye symptom score in the intermittent exotropia (IXT) (red) and control (blue) groups (a) and subgroup of IXT (b).** The red circles and blue crosses indicate individual changes in BFM values and total subjective eye symptom scores (Q1 + Q2 + Q3) in the IXT and control groups. The purple squares and green triangles indicate basic type IXT and convergence insufficiency (CI) type IXT.

−0.025 ± 0.042; *P* = 0.046; Fig 6b). The intercept in BFM was not significantly different between both subgroup (*P* = 0.069).

NPC (0.187 ± 0.245 vs. −0.167 ± 0.461; *P* = 0.51) and fusional vergence range (−0.665 ± 1.586 vs. −3.500 ± 2.638; *P* = 0.40) on total subjective eye symptom score were not significantly different between the patients with basic type IXT and the patients with CI type IXT. The intercept in NPC (*P* = 0.85) and fusional convergence range (*P* = 0.27) on total subjective eye symptom score was not significantly different between the patients with basic type IXT and the patients with CI type IXT.

## Discussion

In the present study, we evaluated the degree of visual fatigue in patients with IXT (basic type and CI type) using objective and subjective indicators. The decrease in BFM was significantly greater in the IXT group than in the control group (Fig 5a). The change in total subjective eye symptom score was also significantly greater in the IXT group than in the control group (Fig 5d). The change in BFM was significantly and negatively correlated with the change in total subjective eye symptom score in the IXT and control groups, respectively (Fig 6a). Moreover, the rate of reduction of BFM with increase in total subjective eye symptom score was significantly greater in the IXT group than in the control group (Fig 6a). These findings have objectively shown that patients with IXT are at a greater risk of visual fatigue in comparison with healthy individuals.

In the present study, the patients and healthy volunteers performed the visual task, in which the target moved reciprocally from 67 cm to 40 cm, with speed 0.5 MA/s under the correction of refractive errors for 5.0 m. All subjects followed the target during the visual task, and vergence and accommodation were recorded simultaneously (Figs 1 and 2). Our findings support the evidence that the vergence and accommodation systems act simultaneously[8]. Moreover, NPC was significantly reduced in the postvisual task than in the previsual task in each group (Figs 3b and 4b). These findings suggest that the present task imposes a strain on vergence.

Binocular coordination in the patient with IXT was gradually disrupted during the visual task (Fig 1a). In contrast, the healthy volunteers maintained binocular coordination during the visual task (Fig 2a). The accommodative response reduced and did not follow the accommodative stimulus after binocular coordination was disrupted in a patient with IXT (Fig 1b). These findings suggest that vergence eye movement in the patients with IXT is fragile as compared with the healthy volunteers; moreover, in patients with IXT, the accommodative response declined due to a lack of convergence accommodation.

The subjective symptom scores of Q1, Q3, Q4, Q5, and Q6 in the IXT group were approximately two-fold higher than those in the control group in the previsual task. (Tables 3 and 4). These subjective symptom scores may relate to the baseline of BFM, NPC, and fusional vergence range because these values were worse in the IXT group than in the control group. The amount of change before and after was evaluated to cancel the carry-over effect in the present study. The changes in the total subjective eye symptom scores were significantly greater in the IXT group than in the control group (Figs 3d–5d and Tables 2 and 3). We have calculated the sum of three symptoms to evaluate visual fatigue because visual fatigue exhibits a variety of symptoms, such as tired eyes, blurry vision, and eye sensation (pain and/or dry eye)[21–23, 27]. In the present study, 11 of the 14 patients were adults with IXT. Von Noorden reported that the frequency of symptoms with visual fatigue was higher in adult patients than in pediatric patients[4]. The physical and psychological discomfort was not significantly different between the IXT and control groups and was not correlated with the BFM, NPC, and fusional

**Table 5. Mean results for the intermittent exotropia (IXT) subgroup.**

| Change in test | Basic type (n = 9) | CI type (n = 5) | *P* value |
|---|---|---|---|
| BFM | −0.200 ± 0.228 | −0.160 ± 0.096 | 0.90 |
| NPC (cm) | 1.0 ± 1.1 | 0.5 ± 1.0 | 0.37 |
| Fusional vergence range (PD) | −3.9 ± 6.8 | −0.2 ± 7.0 | 0.99 |
| Subjective symptom questionnaire | | | |
| Q1 | 1.00 ± 0.71 | 0.60 ± 0.55 | 0.37 |
| Q2 | 0.67 ± 1.00 | 0.20 ± 0.44 | 0.44 |
| Q3 | 0.78 ± 0.67 | 1.20 ± 0.45 | 0.30 |
| Q4 | 0.00 ± 0.50 | 0.00 ± 0.00 | 0.99 |
| Q5 | 0.00 ± 0.71 | 0.00 ± 0.00 | 0.99 |
| Q6 | 0.33 ± 0.86 | 0.00 ± 0.71 | 0.61 |
| Q7 | 0.44 ± 1.13 | 0.20 ± 0.84 | 0.90 |

The error term is the standard deviation. The differences between patient with basic type and CI type were analyzed by the Mann–Whitney U test. CI, convergence insufficiency; BFM, binocular fusion maintenance; NPC, near point of convergence; PD, prism diopter.

vergence range. These findings suggest that the BFM, NPC, and convergence mainly relate to subjective eye symptoms and less relate to subjective physical and psychological discomfort.

BFM is more easily collapsed using binocular stress in patients with IXT than in healthy individuals (Figs 3–5 and Tables 2 and 3). The present findings are in agreement of those reported by Hirota et al.[11], who reported that BFM in the healthy volunteers was significantly reduced and fusional vergence range was not significantly different using binocular stress, suggesting that the BFM test can also be applied to patients with IXT.

Changes in BFM were significantly and negatively correlated with changes in total subjective eye symptom score in both the IXT and the control groups. These findings are consistent with those of Hirota et al. [11], who reported that a change in BFM was significantly and negatively correlated with a change in total subjective eye symptom score in healthy volunteers. Furthermore, the authors have considered that the BFM is a sensitive indicator to evaluate visual fatigue objectively because the control group showed no significant difference between the previsual and postvisual tasks in this study. The rate of reduction of BFM with increase in total subjective eye symptom score (slope) was significantly greater in the IXT group than in the control group, but there was no significant difference between the two groups at the zero point of total subjective eye symptom score (intercept) (Fig 6a). These findings suggest that patients with IXT are at a greater risk of visual fatigue in comparison with healthy individuals.

Although it is preliminary data because the sample size is small, the changes in BFM, NPC, fusional vergence range, and total subjective eye symptom score were not significantly different between the patients with basic type IXT and patients with CI type IXT in the present study (Table 5). However, the slope in BFM on total subjective eye symptom score was significantly and negatively steeper in the basic type IXT than in the CI type IXT (Fig 6b). Thus, we will investigate the difference in type of IXT by increasing the sample size in future work.

## Conclusions

The change in BFM, after a visual task, was significantly lower in the IXT group than in the control group. The change in total subjective eye symptom score, after a visual task, was significantly worse in the IXT group than in the control group. Further, the rate of reduction of BFM with increase in total subjective eye symptom score was significantly greater in the IXT group than in the control group, but there was no significant difference between the two

groups when the subjects were not aware of visual fatigue. These findings using the BFM objectively show that patients with IXT are at a greater risk of visual fatigue in comparison with healthy individuals.

## Supporting information

**S1 Fig. The subjective symptom questionnaire.** Questions 1–3 were designed to assess subjective eye symptoms and Questions 4–7 to assess physical and mental discomfort. The total scores for Q1–3 were used to assess visual fatigue resulting from the visual task. n.p.: no problem.
(DOCX)

**S1 Movie.**
(MP4)

**S1 Table. Distribution for the intermittent exotropia (IXT) group in the previsual task.** The error term is the standard deviation. The normality of previsual task were analyzed by the Shapiro-Wilk test. BFM, binocular fusion maintenance; NPC, near point of convergence; PD, prism diopter.
(DOCX)

**S2 Table. Distribution for the intermittent exotropia (IXT) group in the postvisual task.** The error term is the standard deviation. The normality of postvisual task were analyzed by the Shapiro-Wilk test. BFM, binocular fusion maintenance; NPC, near point of convergence; PD, prism diopter.
(DOCX)

**S3 Table. Distribution for the control group in the previsual task.** The error term is the standard deviation. The normality of postvisual task were analyzed by the Shapiro-Wilk test. BFM, binocular fusion maintenance; NPC, near point of convergence; PD, prism diopter.
(DOCX)

**S4 Table. Distribution for the control group in the postvisual task.** The error term is the standard deviation. The normality of postvisual task were analyzed by the Shapiro-Wilk test. BFM, binocular fusion maintenance; NPC, near point of convergence; PD, prism diopter.
(DOCX)

## Author Contributions

**Conceptualization:** Masakazu Hirota.

**Data curation:** Masakazu Hirota, Kozue Yada.

**Formal analysis:** Masakazu Hirota, Takeshi Morimoto, Takao Endo, Tomomitsu Miyoshi, Takashi Fujikado.

**Funding acquisition:** Masakazu Hirota, Takashi Fujikado.

**Investigation:** Masakazu Hirota.

**Methodology:** Masakazu Hirota.

**Project administration:** Masakazu Hirota, Takashi Fujikado.

**Resources:** Suguru Miyagawa, Yoko Hirohara, Tatsuo Yamaguchi, Makoto Saika.

**Software:** Masakazu Hirota.

**Supervision:** Takashi Fujikado.

**Validation:** Masakazu Hirota.

**Visualization:** Masakazu Hirota.

**Writing – original draft:** Masakazu Hirota.

**Writing – review & editing:** Takashi Fujikado.

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
