## [Decision Letter · Decision Letter 0]

22 Oct 2019

PONE-D-19-23458

Objective Evaluation of Visual Fatigue in Patients with Intermittent Exotropia

PLOS ONE

Dear Dr. Fujikado,

Thank you for submitting your manuscript to PLOS ONE. After careful consideration, we feel that it has merit but does not fully meet PLOS ONE’s publication criteria as it currently stands. Therefore, we invite you to submit a revised version of the manuscript that addresses the points raised during the review process.

Two expert reviewers have evaluated your manuscript. Both reviewers were generally positive, and provided many constructive comments. In particular, Reviewer 1 raises several methodological concerns that need to be addressed, and I particularly agree with the reviewer that the discussion section of the manuscript needs to contextualize the study in the existing literature. Reviewer 2 has additionally provided detailed feedback that will help improve the clarity of the manuscript. Overall I expect it should be possible to address all of the points raised by the reviewers and I look forward to receiving your revised work.

We would appreciate receiving your revised manuscript by Dec 06 2019 11:59PM. To enhance the reproducibility of your results, we recommend that if applicable you deposit your laboratory protocols in protocols.io, where a protocol can be assigned its own identifier (DOI) such that it can be cited independently in the future. For instructions see: http://journals.plos.org/plosone/s/submission-guidelines#loc-laboratory-protocols

We look forward to receiving your revised manuscript.

Kind regards,

Guido Maiello

Academic Editor

PLOS ONE

**Journal Requirements:**

"Masakazu Hirota, Assistant Professor in Teikyo University Faculty of Medical Technology: Patent.

Takeshi Morimoto, Associate Professor in Osaka University Graduate School of Medicine: None.

Takao Endo, Medical Doctor in Osaka University Graduate School of Medicine: None.

Tomomitsu Miyoshi, Assistant Teacher in Osaka University Graduate School of Medicine: None.

Suguru Miyagawa, Employee of Topcon Corporation: Employee (Topcon), Patent.

Yoko Hirohara, Employee of Topcon Corporation: Employee (Topcon).

Tatsuo Yamaguchi, Employee of Topcon Corporation: Employee (Topcon), Patent.

Makoto Saika, Employee of Topcon Corporation: Employee (Topcon), Patent.

Takashi Fujikado, Professor in Osaka University Graduate School of Frontier Biosciences: Financial support (Topcon), Patent."

3.We note that you have stated that you will provide repository information for your data at acceptance. Should your manuscript be accepted for publication, we will hold it until you provide the relevant accession numbers or DOIs necessary to access your data. If you wish to make changes to your Data Availability statement, please describe these changes in your cover letter and we will update your Data Availability statement to reflect the information you provide.

**Comments to the Author**

1. Is the manuscript technically sound, and do the data support the conclusions?

Reviewer #1: Partly

Reviewer #2: Yes

2. Has the statistical analysis been performed appropriately and rigorously? 

Reviewer #1: Yes

Reviewer #2: Yes

3. Have the authors made all data underlying the findings in their manuscript fully available?

Reviewer #1: No

Reviewer #2: Yes

4. Is the manuscript presented in an intelligible fashion and written in standard English?

Reviewer #1: Yes

Reviewer #2: Yes

5. Review Comments to the Author

Reviewer #1: In this research the authors evaluate the degree of fatigue in patients with intermittent exotropia (X(T)). The authors used a binocular fusion maintenance test (BFM) and correlate this measurement (objective measurement) with a subjective eye symptoms score (subjective measurement). They found that the slope was significantly and negatively steeper in the X(T) group than in the control group. From this main finding they conclude that patients with X(T) have a greater risk of visual fatigue compared with control population. This conclusion is in agreement with previous knowledge about X(T).

I find this to be an interesting study although I have a few concerns that I would like to see answered.

-Do the authors have a measurement of test-retest for the subjective questionnaire? For example, a sample of X(T) and controls performing the questionnaire before and after an ordinary visual task like reading. I think that a test-retest for the subjective score should be necessary.

-The authors present the means and the SD in tables and figures, but they use non-parametric statistical tests (like Wilcoxon signed-rank) for comparing measurements. In line 207 they write that this was done after assessment of normality using Shapiro-Wilk but they do not present the numbers. Please include the W value and p-value of the data. Usually, parametric methods are more robust if the data is normally distributed. Have the authors tried to normalize the data using for example the log transformation? Sometimes this log transformation removes the skewness and makes the variances more equal.

-The Table 1 for X(T) is very informative. I missed the table for the control group. I recommend including it. By the way, patient P6 should be considered stereoblind and maybe removed from the study given that his/her stereoacuity is 3019.9 seconds of arc.

-It is a bit surprising (maybe not) that X(T) patients show a value close to 2 in Q1, Q3, Q4 (tired back), and Q5 (tired neck). Thus, even before starting the visual task, X(T) patients are in the middle of the scale (mildly tired). Interestingly, the scores are almost twice as large as controls. I would recommend discuss this result, for example could these starting scores affect BMC, NPC and FVR?

-Line 264-265. The P-value (P=0.139) doesn’t correspond with the P-Value of Table 3 (P=0.002).

-Figure 6 and 7. I am not sure from the information given in the paper. I understand that you are adding the change for Q1 + the change for Q2 + the change for Q3, why don’t use the average of the three changes (a change average)? This way you can also have a standard deviation of the change.

-The Discussion is just a resume of the main findings. The authors should discuss their results with previous studies.

Minor:

-Line 167. “reciprocally”, do you mean “back and forth”?

-Line 170. There is no “Movie 1”.

-Line 531. Replace “with in” for “within”.

-Figure 6b Was the mean calculated taking into account the outlier at 25?

-Figure 6. Line 554: Rewrite “The change in BFM was significantly lower…” The change is greater. You should write it as in your line 284: “ …BFM was significantly and negatively greater…” The same in lines 51 and 355.

-Figure 7. Line 562: “blue circles”. There are no blue circles.

-Figures 4, 5 & 6. Please describe what represent the boxes, i.e. Quartiles, median, 95% confidence intervals, etc.

Reviewer #2: Introduction

65 – delete ‘are’

66-8 It is unclear what this sentence means – consider re-writing.

70 The definition of vergence should be re-considered as ‘movement of opposite eyes’ is not accurate.

76-7 No references have been provided.

Methods

96 – It is not clear how the participants could be age-matched as the age range of patients was 13-60 years and the age range of healthy subjects was 21-51 years. Unless it should say ‘group age-matched?

97 – Where were the healthy subjects recruited from?

103 – ‘all subjects’ – the healthy participants were called subjects and those with IXT were called participants but now referring to them all as subjects. Throughout the paper the terminology is changing between patients, subjects, volunteers, healthy subjects, control subjects – needs to be consistent.

There are insufficient details of BMF and an over-reliance on the reader to refer to the previous paper for details that are better included in this paper. Some further information that may be helpful include:

BWFA measures binocular eye movements but how does that measure/relate to BMF?

Were the binocular eye movements used to calculate vergence?

What are the variable liquid crystal shutters for?

Was the transmittance changed in both eyes or just in the non-dominant eye as in the previous paper?

Why is the transmission altered?

149-62 – It is not clear why NPC and fusional vergence range are being measured and analysed. The aim of the study suggests that BMF and symptoms will be analysed but so far there has been no mention of these tests and their relevance.

171 – How long did the visual task last?

174 – ‘the subjects’ implies the controls only, based on the earlier section but I assume the patients completed this too. I’d suggest referring to them as patients and controls and then all subjects when referring to them collectively.

176 – According to the previous paper this questionnaire has been taken from elsewhere and therefore a reference is required.

177 – It would be useful to state what questions 1-3 are.

178 – Were the results of questions 4-7 discarded as they are not relevant?

179 – Were the results from the questionnaire totalled or averaged?

199 – It is not clear what the binocular fusion break time is.

209 Unclear what is meant by ‘assess differences in the visual tasks’

Results

234 – What is ‘bi’? Assume this should be ‘BI’ for base in?

248 – There was no mention of accommodation in the method section. How and when was this measured?

250 – Is this an example from an individual control and patient? Need to clearly state this. Do all others follow these 2 examples?

251-2 How was it determined if binocular coordination was maintained or disrupted? Just by eye-balling the data? Fig 2 should be stated as Fig 2a and 2b within the text.

257 – ‘prolonged’ – consider choice of word. Perhaps reduced or worse is more appropriate?

261 – Assume a greater score means worse symptoms but it is unclear why the eye symptom scores would be greater before performing the visual task.

264 – state that the change in BFM was not significant yet the table shows a significant difference.

271 – should state that Q2-3 were not significant.

Table 2:

Contrary to the text, the symptom scores are worse post task.

What was the purpose of analysing Q4-7 as they do not seem relevant?

It would be useful to include asterisks to highlight significant p values.

289 – It is unclear which questions are included in ‘total subjective eye symptom score’.

298 – Interesting that the NPC and fusion range are not correlated with symptom score – yet this isn’t covered later in the discussion.

Table 4: Change in BFM but are the other tests also change from pre- to post- task?

Discussion

350-2 The NPC was significantly reduced in both groups after the visual task but accommodation was only reduced in the IXT group so it is difficult to follow this sentence.

352-3 There was no mention of this link between accommodation and dominance in the results section. Is this just based on the single example given?

355 – ‘changes in BFM were significantly lower in the IXT group than in the control group’ implies that there is less change in the IXT group i.e. less fatigued by the visual task.

365-6 – what 2 factors? There has been no previous mention of the impact of age. Not clear what point is being made.

368 – ‘eye feeling’?

373 – Why is there a significant correlation in the control group?

Conclusion

390-2 Change after a visual task?

395 – what is meant by ‘zero point’?

396-7 – Using the BFM?

Figure 1: It would be better to combine figure 1 with figures 2 and 3 to show the ideal response. The figures in 2 and 3 would be easier to interpret if it could be seen what the expected response should be.

Figures 2 and 3 – What is the purpose of showing all 4 trials? Could they display the average response? That way it could include one figure for the patient with IXT, which displays the ideal response (taken from figure 1), the average vergence and the average accommodation. Another figure could show the same detail for the control subject.

531 – ‘vergence range values with in the with IXT group’

All figures:

Keys would be useful.

Legends contain too much detail – e.g. methods are included and the analysis performed.

6. PLOS authors have the option to publish the peer review history of their article (what does this mean?). If published, this will include your full peer review and any attached files.

Reviewer #1: No

Reviewer #2: No

---

## [Author Response · Author response to Decision Letter 0]

7 Dec 2019

Reviewer #1: In this research the authors evaluate the degree of fatigue in patients with intermittent exotropia (X(T)). The authors used a binocular fusion maintenance test (BFM) and correlate this measurement (objective measurement) with a subjective eye symptoms score (subjective measurement). They found that the slope was significantly and negatively steeper in the X(T) group than in the control group. From this main finding they conclude that patients with X(T) have a greater risk of visual fatigue compared with control population. This conclusion is in agreement with previous knowledge about X(T).

I find this to be an interesting study although I have a few concerns that I would like to see answered.

-Do the authors have a measurement of test-retest for the subjective questionnaire? For example, a sample of X(T) and controls performing the questionnaire before and after an ordinary visual task like reading. I think that a test-retest for the subjective score should be necessary.

Thank you for your constructive comments. My colleagues and I have assented to your comment and revised our manuscript.

We have measured the repeatability of the subjective questionnaire in the healthy volunteers and patients with intermittent exotropia (IXT) in the pilot study. We recruited three healthy individuals and three patients with IXT. They have been performed the binocular fusion maintenance test and before and after the same visual task this experiment twice at the same time on another day. 

Total subjective eye symptom scores were higher in the post-visual task than in the pre-visual task both the first test and second test in healthy individuals and patients with IXT (Appendix Figure 1).

Appendix Figure 1. Repeatability of total subjective eye symptom score in the healthy individuals (a) and patients with IXT (b)

The blue and red boxplots with dots indicate the pre- and post-visual task. Total subjective eye symptom scores were higher in the post-visual task than in the pre-visual task both the first test and second test in healthy individuals and patients with IXT.

The authors present the means and the SD in tables and figures, but they use non-parametric statistical tests (like Wilcoxon signed-rank) for comparing measurements. In line 207 they write that this was done after assessment of normality using Shapiro-Wilk but they do not present the numbers. Please include the W value and p-value of the data. Usually, parametric methods are more robust if the data is normally distributed. Have the authors tried to normalize the data using for example the log transformation? Sometimes this log transformation removes the skewness and makes the variances more equal.

We agree with your comments. We consider that the non-parametric data should not show SD, but should show IQR. However, the readers can make a sense SD better than IQR. Thus, we used the boxplot with dots to show the non-normal distribution and described SD data at the main text and table.

We agree with your point out. We have omitted the values of Shapiro-Wilk test because of the statistical analysis to use the normality test before the comparative test. We created a supplementary table for W- and P-values of at the Results section.

The data with non-normally distribution transform to logarithms is one of the statistical techniques to close the normal distribution spuriously. We have already tried the log transform. However, data have not improved. 

The Table 1 for X(T) is very informative. I missed the table for the control group. I recommend including it. By the way, patient P6 should be considered stereoblind and maybe removed from the study given that his/her stereoacuity is 3019.9 seconds of arc.

We added a table for the control group. We excluded patient 6. 

-It is a bit surprising (maybe not) that X(T) patients show a value close to 2 in Q1, Q3, Q4 (tired back), and Q5 (tired neck). Thus, even before starting the visual task, X(T) patients are in the middle of the scale (mildly tired). Interestingly, the scores are almost twice as large as controls. I would recommend discuss this result, for example could these starting scores affect BMC, NPC and FVR?

Thank you for your constructive comments. We described this in Discussion section.

The subjective symptom scores of Q1, Q3, Q4, Q5, and Q6 in the IXT group were about twice as high compared with the control group at the starting the visual task (Tables 3 and 4). These subjective symptom scores may relate to the baseline of BFM, NPC, fusional vergence range because these values were worse in the IXT group than in the control group. The amount of change before and after was evaluated to cancel the carry-over effect in the present study.

-Line 264-265. The P-value (P=0.139) doesn’t correspond with the P-Value of Table 3 (P=0.002).

We fixed the P-value. 

-Figure 6 and 7. I am not sure from the information given in the paper. I understand that you are adding the change for Q1 + the change for Q2 + the change for Q3, why don’t use the average of the three changes (a change average)? This way you can also have a standard deviation of the change.

We have already presented individual mean and standard deviations in Tables. Furthermore, the total subjective eye symptom score becomes the mean of mean, If the individual three scores were averaged. Thus, we used the adding of three values. Although the slopes coefficient is different to use the mean value, the p-value does not change.

-The Discussion is just a resume of the main findings. The authors should discuss their results with previous studies.

We revised the discussion section. 

Minor:

-Line 167. “reciprocally”, do you mean “back and forth”?

Yes. We added the word from “reciprocally” to “reciprocally (back and forth)”.

-Line 170. There is no “Movie 1”.

I am very sorry. We had not attached Movie 1 in the first submission. We attach Movie 1 in the revision.

-Line 531. Replace “with in” for “within”.

We fixed the word from “with in” to “within”.

-Figure 6b Was the mean calculated taking into account the outlier at 25?

No. The mean value calculated arithmetic mean. 

-Figure 6. Line 554: Rewrite “The change in BFM was significantly lower…” The change is greater. You should write it as in your line 284: “ …BFM was significantly and negatively greater…” The same in lines 51 and 355.

We rewrote the sentence in Figure 6 from “The change in BFM was significantly lower…” to “The change in BFM was significantly and negatively greater”. 

-Figure 7. Line 562: “blue circles”. There are no blue circles.

We typo the sentence. “The red circles and blue crosses” is correct. 

-Figures 4, 5 & 6. Please describe what represent the boxes, i.e. Quartiles, median, 95% confidence intervals, etc.

We added the information of quartiles, median, and 95% confidence interval in the Figures 4, 5, and 6.

Reviewer #2: Introduction

Thank you for your constructive comments. My colleagues and I have assented to your comment and revised our manuscript.

65 – delete ‘are’

We deleted ‘are’. 

66-8 It is unclear what this sentence means – consider re-writing.

We rewrote the sentence. 

Earlier questionnaire survey studies revealed that visual fatigue is one of the most prevalent symptoms in patients with IXT.

70 The definition of vergence should be re-considered as ‘movement of opposite eyes’ is not accurate.

We modified the sentence for vergence. 

Vergence is the simultaneous movement to align both eyes for obtain or maintain binocular vision…

76-7 No references have been provided.

We mentioned the references. 

Methods

96 – It is not clear how the participants could be age-matched as the age range of patients was 13-60 years and the age range of healthy subjects was 21-51 years. Unless it should say ‘group age-matched?

We fixed the sentence from ‘age-matched’ to ‘group age-matched’ in the explanation of control group. 

97 – Where were the healthy subjects recruited from?

We recruited subjects using online recruitment. 

103 – ‘all subjects’ – the healthy participants were called subjects and those with IXT were called participants but now referring to them all as subjects. Throughout the paper the terminology is changing between patients, subjects, volunteers, healthy subjects, control subjects – needs to be consistent.

We consist the terminology: from ‘control subjects’ to ‘healthy volunteers’. ‘Subjects’ use to all participants who include patients and healthy volunteers.

There are insufficient details of BMF and an over-reliance on the reader to refer to the previous paper for details that are better included in this paper. Some further information that may be helpful include:

BWFA measures binocular eye movements but how does that measure/relate to BMF?

We added some explanations for details of BFM test.

BFM can be assessed by reducing the intensity of incident light on one eye, which is defined by the number of photons, because the perceptive size of retinal image depends on the intensity of incident light[17]. Moreover, the binocular fusion break can be judged automatically to record the eye movements because one eye deviate exo- or eso-direction after the binocular fusion break[18].

Were the binocular eye movements used to calculate vergence?

Yes, it was. We mentioned this information.

The binocular eye movements used to calculate vergence.

What are the variable liquid crystal shutters for?

I’m sorry. Variable liquid crystal shutters are not exist. We fixed the sentence from “Variable liquid crystal shutters” to “The liquid crystal shutters”.

Was the transmittance changed in both eyes or just in the non-dominant eye as in the previous paper?

The transmittance changed just in nondominant eye. We mentioned the “just in nondominant eye”.

The transmittance of the liquid crystal shutter changed just in the nondominant eye, which was determined by a hole-in-the-card test, was set at 23.0% for 2 s and was then reduced sequentially by 1.15% every second.

Why is the transmission altered?

BFM can be assessed by reducing the intensity of incident light on one eye, which is defined by the number of photons, because the perceptive size of retinal image depends on the intensity of incident light. We described the first paragraph of Binocular Fusion Maintenance section with your constructive comment.

149-62 – It is not clear why NPC and fusional vergence range are being measured and analysed. The aim of the study suggests that BMF and symptoms will be analysed but so far there has been no mention of these tests and their relevance.

We used the target moved back and forth from 67 cm to 40 cm. We had considered the vergence abilities significantly reduced after the visual task at the experimental protocol creation stage in this study. We described that why both tests performed in this study at the Method section.

IXT is characterized by prolonged NPC and low fusional convergence at close distances. Thus, all subjects performed the NPC and fusional vergence range test at near before and after the visual task.

171 – How long did the visual task last?

The visual task completed within 2 min.

174 – ‘the subjects’ implies the controls only, based on the earlier section but I assume the patients completed this too. I’d suggest referring to them as patients and controls and then all subjects when referring to them collectively.

We consist the terminology: from ‘control subjects’ to ‘healthy volunteers’. ‘Subjects’ use to all participants who include patients and healthy volunteers.

176 – According to the previous paper this questionnaire has been taken from elsewhere and therefore a reference is required.

We added the references of Nakazawa et al., Sheedy and Bergstrom, and Hoffman et al.

177 – It would be useful to state what questions 1-3 are.

We added the information for Questions 1–3. 

Questions 1–3 (1, How tired are your eyes?; 2, How clear is your vision?; 3, How do your eyes feel?) were designed to assess subjective eye symptoms.

178 – Were the results of questions 4-7 discarded as they are not relevant?

The physical and psychological discomfort was one of the secondary endpoints. Moreover, we considered that the subjects might be have a psychological bias if the questionnaire evaluates only visual fatigue.T

Questions 1–3 (1, How tired are your eyes?; 2, How clear is your vision?; 3, How do your eyes feel?) were designed to assess subjective eye symptoms.

179 – Were the results from the questionnaire totalled or averaged?

We used totaled score. We mentioned the totaled score at subjective symptoms questionnaire section.

The subjective eye symptom scores (Q1, Q2, Q3) were totaled as total subjective eye symptom score.

199 – It is not clear what the binocular fusion break time is.

We described the detail for calculation of binocular fusion break.

The eye position data collected during the 50 s measurement periods were averaged over the three trials before and after the visual task. The binocular fusion break time (TB) was calculated automatically from the nondominant eye movements based on the results of our previous study using Python 3.6.5. Basemin was determined as the average eye position over 2 seconds after the beginning of measurement in which the transmittance of the liquid crystal shutter remained equal between the right and the left eye. Basemax was determined by the average eye position in the nondominant eye over 2 seconds between 25 and 27 seconds in which the difference of transmittance between the right and left eye was the largest. The amplitude in the deviation of the nondominant eye (Dn) was calculated as [Basemax – Basemin]. Then, we determined the points in which the amplitude of deviation in the non-dominant eye reached 10% and 90% of the total amplitude as 0.1Dn and 0.9Dn respectively during the fusion break phase. A linear regression line was created using the nondominant eye position at 0.1 Dn and 0.9 Dn. We then determined TB as the intersection between Basemin and the linear regression line of the fusion break phase.

209 Unclear what is meant by ‘assess differences in the visual tasks’

We deleted the sentences of “To assess differences in the visual tasks”.

To assess the significance of the differences between the IXT and control groups in the changes (post − pre) in BFM, NPC, fusional vergence range, and total subjective eye symptom scores (Q1 + Q2 + Q3) was determined by the Mann–Whitney U test after assessment of normality by the Shapiro–Wilk test.

Results

234 – What is ‘bi’? Assume this should be ‘BI’ for base in?

I’m sorry. ‘bi” indicate “base-in” as your pointed out. However, the minus sign indicates exodeviation in this study. We mentioned this at the Subjects section. 

248 – There was no mention of accommodation in the method section. How and when was this measured?

The binocular wavefront aberrometer can measure the accommodation simultaneously. We described that all subjects recorded the binocular eye movements and accommodative responses at the visual task section. 

All subjects underwent four trials, and recorded the binocular eye movements and accommodative responses simultaneously.

250 – Is this an example from an individual control and patient? Need to clearly state this. Do all others follow these 2 examples?

We displayed patient 10 and healthy volunteer 14. Almost all patients and healthy volunteers showed similar to the representative data.

251-2 How was it determined if binocular coordination was maintained or disrupted? Just by eye-balling the data? Fig 2 should be stated as Fig 2a and 2b within the text.

The binocular coordination was checked the numerical data and anterior video data. We fixed the notation from “Fig 2” to “Fig 2a and Fig 2b”. 

257 – ‘prolonged’ – consider choice of word. Perhaps reduced or worse is more appropriate?

Thank you for your recommendation. We changed the word from “prolonged” to “reduced”. 

261 – Assume a greater score means worse symptoms but it is unclear why the eye symptom scores would be greater before performing the visual task.

We changed the word from “greater” to “worse”. The score was significantly worse in the postvisual task than in the previsual task. We fixed the incorrect information.

Subjective eye symptom scores (Q1, Q2, and Q3) were significantly worse in the postvisual task than in the previsual task (Q1, P = 0.003; Q2, P = 0.034; Q3, P = 0.002; Table 3, Supplementary Table 1 and 2).

The subjective eye symptom score (Q1) was significantly worse in the postvisual task than in the previsual task (Q1, P = 0.013; Table 4, Supplementary Table 3 and 4).

264 – state that the change in BFM was not significant yet the table shows a significant difference.

I’m very sorry. The state on the main text is correct. We fixed the information on BFM in Table 4.

271 – should state that Q2-3 were not significant.

We stated that Q2 and Q3 were not significant in the control group.

The subjective eye symptom scores of Q2 (pre, 0.87 ± 0.64; post, 0.80 ± 0.41; P = 0.57; Table 4) and Q3 (pre, 1.13 ± 0.74; post, 1.46 ± 0.64; P = 0.096; Table 4) were not significantly different between the previsual and postvisual task.

Table 2:

Contrary to the text, the symptom scores are worse post task.

What was the purpose of analysing Q4-7 as they do not seem relevant?

It would be useful to include asterisks to highlight significant p values.

The state on the main text is correct. We added the asterisks for the remark of the significant difference in Tables 3 and 4.

289 – It is unclear which questions are included in ‘total subjective eye symptom score’.

We mentioned the detail of Q1, Q2, and Q3. Then, we also described the total subjective eye symptom score defined that totaled Q1, Q2, and Q3 in Subjective Symptoms Questionnaire section.

298 – Interesting that the NPC and fusion range are not correlated with symptom score – yet this isn’t covered later in the discussion.

We covered NPC and fusional range in the discussion.

Table 4: Change in BFM but are the other tests also change from pre- to post- task?

Yes, it was. We described the change of NPC.

Discussion

350-2 The NPC was significantly reduced in both groups after the visual task but accommodation was only reduced in the IXT group so it is difficult to follow this sentence.

We consider that the present task imposes a strain on vergence because NPCs in both groups were significantly reduced after the visual task. Also, binocular coordination was not gradually disrupted in the healthy volunteer, but in the patients with IXT. We have considered that vergence eye movement in the patients with IXT is fragile comparison with the healthy volunteers, and the patients with IXT the accommodative response declined because of the lack of convergence accommodation.

352-3 There was no mention of this link between accommodation and dominance in the results section. Is this just based on the single example given?

I’m sorry. We mentioned that all patients with IXT followed the target using the dominant eye in the Result section. Also, we deleted the sentences that accommodative response in the nondominant eye was driven by the dominant eye.

355 – ‘changes in BFM were significantly lower in the IXT group than in the control group’ implies that there is less change in the IXT group i.e. less fatigued by the visual task.

We fixed the word from “lower” to “worse”.

The changes in BFM were significantly worse in the IXT group than in the control group, although the fusional vergence range was not significantly different (Figs. 4 – 6; Table 2 and 3).

365-6 – what 2 factors? There has been no previous mention of the impact of age. Not clear what point is being made.

I’m sorry. The impact of age was none. We deleted this sentence.

368 – ‘eye feeling’?

We use “eye feeling” means “pain” and/or “dry eye”. 

373 – Why is there a significant correlation in the control group?

We consider that the BFM is a sensitive indicator to evaluate visual fatigue objectively because the control group did not significantly different between pre- and postvisual tasks in this study.

Conclusion

390-2 Change after a visual task?

We modified the sentence from “The change in BFM…” to “The change after a visual task in BFM…”.

395 – what is meant by ‘zero point’?

Zero point means none. We fixed the words from “zero point” to “none”.

396-7 – Using the BFM?

We modified the sentence from “These findings objectively show …” to “These findings using the BFM objectively show…”.

Figure 1: It would be better to combine figure 1 with figures 2 and 3 to show the ideal response. The figures in 2 and 3 would be easier to interpret if it could be seen what the expected response should be.

We modified and merged Figure 1 on figures 2 and 3.

Figures 2 and 3 – What is the purpose of showing all 4 trials? Could they display the average response? That way it could include one figure for the patient with IXT, which displays the ideal response (taken from figure 1), the average vergence and the average accommodation. Another figure could show the same detail for the control subject.

We displayed that the patients with IXT were gradually disrupted from trial 1 to 4. The average response also worse in the patients with IXT than in the healthy volunteers as you point out. We modified the Figure 2 (patient) and 3 (healthy volunteer).

531 – ‘vergence range values with in the with IXT group’

We fixed the sentence from “vergence range values with in the with IXT group” to “vergence range values within the IXT group”.

All figures:

Keys would be useful.

Legends contain too much detail – e.g. methods are included and the analysis performed.

We deleted lengthy sentences in the figure legends.

---

## [Decision Letter · Decision Letter 1]

10 Jan 2020

PONE-D-19-23458R1

Objective Evaluation of Visual Fatigue in Patients with Intermittent Exotropia

PLOS ONE

Dear Dr. Fujikado,

Thank you for submitting your manuscript to PLOS ONE. After careful consideration, we feel that it has merit but does not fully meet PLOS ONE’s publication criteria as it currently stands. Therefore, we invite you to submit a revised version of the manuscript that addresses the points raised during the review process.

Reviewer 2 has some additional detailed and constructive suggestions on how to improve the clarity of the methods section that would greatly benefit the manuscript. Referencing Movie 1 may help address some of the reviewer comments, but please make sure to rename the movie file correctly so that readers are not confused as to what you are referring to. The reviewer comments should all be straightforward to address and I look forward to receiving your revised work.

We would appreciate receiving your revised manuscript by Feb 24 2020 11:59PM. To enhance the reproducibility of your results, we recommend that if applicable you deposit your laboratory protocols in protocols.io, where a protocol can be assigned its own identifier (DOI) such that it can be cited independently in the future. For instructions see: http://journals.plos.org/plosone/s/submission-guidelines#loc-laboratory-protocols

We look forward to receiving your revised manuscript.

Kind regards,

Guido Maiello

Academic Editor

PLOS ONE

Reviewers' comments:

Reviewer's Responses to Questions

**Comments to the Author**

1. If the authors have adequately addressed your comments raised in a previous round of review and you feel that this manuscript is now acceptable for publication, you may indicate that here to bypass the “Comments to the Author” section, enter your conflict of interest statement in the “Confidential to Editor” section, and submit your "Accept" recommendation.

Reviewer #1: All comments have been addressed

Reviewer #2: (No Response)

2. Is the manuscript technically sound, and do the data support the conclusions?

Reviewer #1: Yes

Reviewer #2: Yes

3. Has the statistical analysis been performed appropriately and rigorously? 

Reviewer #1: Yes

Reviewer #2: Yes

4. Have the authors made all data underlying the findings in their manuscript fully available?

Reviewer #1: No

Reviewer #2: Yes

5. Is the manuscript presented in an intelligible fashion and written in standard English?

Reviewer #1: Yes

Reviewer #2: Yes

6. Review Comments to the Author

Reviewer #1: The authors answer all my questions. By the way, I couldn't find the Movie 1 in this version either.

Reviewer #2: Thank you for taking the time to address the previous comments. However, I still have some concerns, especially regarding the methodology, that need addressing.

The line numbers refer to the track changes version document.

The procedure of measuring BFM is not clear. Two new sentences have been added from line 116. The second sentence would be better placed at the end of that p/g, on line 127. The second sentence is also unclear: ‘fusion break can be judged automatically to record the eye movements because one eye deviates’. The underlined text does not make sense. Is the fusion break determined by the equipment or by the examiner viewing the eyes, looking for one eye to deviate?

In my previous comments I had asked what liquid crystal shutters are but this has not been addressed. Please explain what these are for? Are they used to reduce light intensity?

145 Is this target viewed on the plate or a screen?

151 Remove ‘changed just in’ as the last part of the sentence does not follow with this included. It is better without.

149-58 It is not clear what target the subjects are looking at during this procedure. Is it the same one that was used to correct refractive errors.

It is still not clearly stated what you are measuring in the BFM test. Is it the intensity of light when then fusion breaks? What are the units? This is important to include and could be added on line 157.

184 How was the target moved back and forth? Was this on a motorised beam?

189 All subjects underwent four trials – what is a trial? Do you mean 4 x the visual task? The motion was performed 3 times, so in total 12 motions, before all the measurements were repeated?

189 Make it clearer that the eye movements and accommodation were measured during the visual task.

190 Accommodation has not been previously mentioned. A response to my previous comment explains the wavefront aberrometer measures this. This needs to be included earlier on 149 where it states what measurements are taken.

199 Unclear justification provided to previous comment on why Q4-7 need to be included. These are not significant anyway. If keeping them in, then you should state somewhere what the questions were. Q1-3 have now been included in the text so the same could be done with these. Alternatively, include the questions in a table.

291 Would it be better to present the healthy volunteer results first, in figure 1? And the IXT results in figure 2?

In my previous comments I had asked how you determined if binocular coordination had been disrupted. This was answered in the response document but has not been included in the manuscript. It would be helpful to the reader, as it was assumed this was done by eyeballing the data.

Figure 4 and 5 legends still have results included. Results should be included in the main manuscript and not in the legend.

Table 4. You appeared to have misread by previous comment. I had asked if NPC and fusional vergence was also change in values from pre- to post- task, like the BFM and survey. You had answered yes but did not change the table accordingly. I would suggest changing ‘Test’ to ‘Change in test’ or ‘change in test result’? That way it is clear that all the tests listed are looking at the change. Remove ‘change’ in front of BFM and subjective symptom questionnaire as the new heading now makes this clear.

422 It is not clear how or why Q4-7 would related to BFM, NPC scores etc.

432 It is stated that in a previous study BFM was significantly decreased but it is not stated who was being tested. What condition did they have or were they controls?

435 Consider the structure. Already covered survey questions in an earlier p/g, then moved onto BFM, and now coming back to survey questions. The next p/g moves onto discussing BFM again.

467 You’ve now made it clear that the change is related to after performing the visual task but this information is not presented in the correct part of the sentence. ‘The change after a visual task in BFM was significantly lower’ should actually be ‘The change in BFM, after a visual task, was significantly lower’. This should be changed in the next sentence too.

473 I had commented that ‘at the zero point’ was not clear. This has been changed to ‘none’ but the sentence still does not make sense. I don’t understand what point is being made. Revise this sentence.

Carefully proof read the manuscript as there are still many typographical and grammatical errors. Here are some of them:

- 74 ‘to align both eyes for obtain’ should be ‘to align both eyes to obtain’

- 140 ‘eye movements used to calculate’ should be ‘eye movements were used to calculate’

- 231 avoid ‘we’/first person

- 298 ‘showed similar to the representative data’. Do you mean showed similar results?

- 398 ‘vergence and accommodation working simultaneously’. Do you mean ‘vergence and accommodation were recorded simultaneously’?

- 420 ‘at the starting visual task’ needs re-writing

- 450 ‘did not significantly different’ should be ‘did not significantly differ’.

7. PLOS authors have the option to publish the peer review history of their article (what does this mean?). If published, this will include your full peer review and any attached files.

Reviewer #1: No

Reviewer #2: No

---

## [Author Response · Author response to Decision Letter 1]

16 Feb 2020

Reviewer #1: The authors answer all my questions. By the way, I couldn't find the Movie 1 in this version either.

Thank you for the re-review. My colleagues and I were able to improve the manuscript thanks to your constructive comments.

I uploaded a movie file in the file inventory at the time of the first revision (Photo 1). I have carefully confirmed that the movie is visible. You can see the video by clicking the pdf of Movie1.

Reviewer #2: Thank you for taking the time to address the previous comments. However, I still have some concerns, especially regarding the methodology, that need addressing.

The line numbers refer to the track changes version document.

The procedure of measuring BFM is not clear. Two new sentences have been added from line 116. The second sentence would be better placed at the end of that p/g, on line 127. The second sentence is also unclear: ‘fusion break can be judged automatically to record the eye movements because one eye deviates’. The underlined text does not make sense. Is the fusion break determined by the equipment or by the examiner viewing the eyes, looking for one eye to deviate?

In my previous comments I had asked what liquid crystal shutters are but this has not been addressed. Please explain what these are for? Are they used to reduce light intensity?

Thank you for your constructive comments. We have revised our manuscript in accordance to your comments. The line numbers refer to the track changes version document.

The second sentence has been moved from line 133 to line 135. We have used the word “automatically” to mean objectively evaluating with the BWFA. 

We would like to apologize for not providing an explanation. We used liquid crystal shutters to reduce light intensity (line 137 – 138).

145 Is this target viewed on the plate or a screen?

We used a printed plate and not a digital screen (line 158).

151 Remove ‘changed just in’ as the last part of the sentence does not follow with this included. It is better without.

We have removed the words “changed just in” (line 166).

149-58 It is not clear what target the subjects are looking at during this procedure. Is it the same one that was used to correct refractive errors.

It is still not clearly stated what you are measuring in the BFM test. Is it the intensity of light when then fusion breaks? What are the units? This is important to include and could be added on line 157.

We have modified the sentences to explain the procedure. 

The fixation target used was same as that the one used to correct refractive errors (lines 162 – 163).

The BFM test evaluated the intensity of incident light ratio with both eyes when binocular fusion breaks (lines 175–176).

184 How was the target moved back and forth? Was this on a motorized beam?

We moved the target using an electric motor (line 202).

189 All subjects underwent four trials – what is a trial? Do you mean 4 x the visual task? The motion was performed 3 times, so in total 12 motions, before all the measurements were repeated?

We defined the 1 trial as three reciprocating motions (lines 202–203). The subjects underwent 4 trials, so 12 motions in total. 

189 Make it clearer that the eye movements and accommodation were measured during the visual task.

We have modified the sentences; eye movements and accommodative responses were measured simultaneously during the visual task.

190 Accommodation has not been previously mentioned. A response to my previous comment explains the wavefront aberrometer measures this. This needs to be included earlier on 149 where it states what measurements are taken.

We have mentioned that we measured the wavefront aberrations of second orders (accommodative response) (line 164).

199 Unclear justification provided to previous comment on why Q4-7 need to be included. These are not significant anyway. If keeping them in, then you should state somewhere what the questions were. Q1-3 have now been included in the text so the same could be done with these. Alternatively, include the questions in a table.

We apologize for the misunderstanding. We used Questions 4–7 to avoid bias. As you pointed out, Questions 4–7 are less important than Questions 1–3. We described the contents of Questions 4–7 in the main text (lines 224–233).

291 Would it be better to present the healthy volunteer results first, in figure 1? And the IXT results in figure 2?

In my previous comments I had asked how you determined if binocular coordination had been disrupted. This was answered in the response document but has not been included in the manuscript. It would be helpful to the reader, as it was assumed this was done by eyeballing the data.

We have changed the figure number of 1 and 2. The healthy volunteer results are shown in Figure 1, whereas IXT results have been shown in Figure 2.

We have mentioned about checking of binocular coordination from the numerical data and anterior video data (lined 357–358).

Figure 4 and 5 legends still have results included. Results should be included in the main manuscript and not in the legend.

The data of quartiles, medians, and 95% confidence intervals have neem moved from the legends to the main text.

Table 4. You appeared to have misread by previous comment. I had asked if NPC and fusional vergence was also change in values from pre- to post- task, like the BFM and survey. You had answered yes but did not change the table accordingly. I would suggest changing ‘Test’ to ‘Change in test’ or ‘change in test result’? Th at way it is clear that all the tests listed are looking at the change. Remove ‘change’ in front of BFM and subjective symptom questionnaire as the new heading now makes this clear.

We have misread your comment in the first revision. The subgroup in IXT did not find significantly differ in other parameters. We have changed the table number from 4 to 5 and have also changed the columns from “Test” to “Change in test.”

422 It is not clear how or why Q4-7 would related to BFM, NPC scores etc.

We added the analysis for total subjective physical and psychological discomfort (line 304 – 305, 309 – 310, 513 – 514, 537 – 548).

432 It is stated that in a previous study BFM was significantly decreased but it is not stated who was being tested. What condition did they have or were they controls?

We mentioned that the previous study evaluated the BFM in healthy volunteers (line 640).

435 Consider the structure. Already covered survey questions in an earlier p/g, then moved onto BFM, and now coming back to survey questions. The next p/g moves onto discussing BFM again.

We have modified the structure. This paragraph has been merged with the fourth paragraph of the Discussion.

467 You’ve now made it clear that the change is related to after performing the visual task but this information is not presented in the correct part of the sentence. ‘The change after a visual task in BFM was significantly lower’ should actually be ‘The change in BFM, after a visual task, was significantly lower’. This should be changed in the next sentence too.

We have revised the sentence as per your comments (lines 722–724). 

473 I had commented that ‘at the zero point’ was not clear. This has been changed to ‘none’ but the sentence still does not make sense. I don’t understand what point is being made. Revise this sentence.

The phrase “At the zero point” means “the subjects were not aware of visual fatigue.” We have mentioned this in the Conclusion section (lines 727 – 728).

Carefully proof read the manuscript as there are still many typographical and grammatical errors. Here are some of them:

We have carefully proofread our manuscript. Furthermore, the manuscript has been proofread by native English speakers. 

- 74 ‘to align both eyes for obtain’ should be ‘to align both eyes to obtain’

We have revised the word “for obtain" to “to obtain” (line 70).

- 140 ‘eye movements used to calculate’ should be ‘eye movements were used to calculate’

We have added the word “were” (line153).

- 231 avoid ‘we’/first person

We have avoided the use of first person (lines 276).

- 298 ‘showed similar to the representative data’. Do you mean showed similar results?

Yes, it does. All healthy volunteers maintained binocular coordination during the visual task. In all patients with IXT, binocular coordination was gradually disrupted during the visual task (lines 368 – 371).

- 398 ‘vergence and accommodation working simultaneously’. Do you mean ‘vergence and accommodation were recorded simultaneously’?

Yes, it does. I have revised “working” to “recorded” (line 599).

- 420 ‘at the starting visual task’ needs re-writing

As per your suggestion, we have rewritten the sentence (lines 614 – 616).

- 450 ‘did not significantly different’ should be ‘did not significantly differ’.

We have revised the word “different” to “differ” (line 686).

---

## [Decision Letter · Decision Letter 2]

10 Mar 2020

Objective Evaluation of Visual Fatigue in Patients with Intermittent Exotropia

PONE-D-19-23458R2

Dear Dr. Fujikado,

We are pleased to inform you that your manuscript has been judged scientifically suitable for publication and will be formally accepted for publication once it complies with all outstanding technical requirements.

With kind regards,

Guido Maiello

Academic Editor

PLOS ONE

Additional Editor Comments (optional):

Reviewers' comments:

Reviewer's Responses to Questions

**Comments to the Author**

1. If the authors have adequately addressed your comments raised in a previous round of review and you feel that this manuscript is now acceptable for publication, you may indicate that here to bypass the “Comments to the Author” section, enter your conflict of interest statement in the “Confidential to Editor” section, and submit your "Accept" recommendation.

Reviewer #2: All comments have been addressed

2. Is the manuscript technically sound, and do the data support the conclusions?

Reviewer #2: (No Response)

3. Has the statistical analysis been performed appropriately and rigorously? 

Reviewer #2: (No Response)

4. Have the authors made all data underlying the findings in their manuscript fully available?

Reviewer #2: (No Response)

5. Is the manuscript presented in an intelligible fashion and written in standard English?

Reviewer #2: (No Response)

6. Review Comments to the Author

Reviewer #2: Thank you for editing your article as requested. I am happy with the changes and feel the manuscript is ready for submission.

If possible, it would be appreciated if one more minor change could be made. It was previously unclear what the target was and whether the same target was used for the other tests. Thank you for making the appropriate changes to amend this. Since you have now stated on line 142 what the target is, I think the sentence starting on 146 can now be cut down from 'The subjects continued to fixate the starburst target that was same as the one used to correcting the refractive errors' to 'The subjects continued to fixate the starburst target', as that makes it clear enough that the same target was used.

Thanks

7. PLOS authors have the option to publish the peer review history of their article (what does this mean?). If published, this will include your full peer review and any attached files.

Reviewer #2: No

---

## [Editor Report · Acceptance letter]

12 Mar 2020

PONE-D-19-23458R2 

Objective Evaluation of Visual Fatigue in Patients with Intermittent Exotropia 

Dear Dr. Fujikado:

I am pleased to inform you that your manuscript has been deemed suitable for publication in PLOS ONE. Congratulations! Your manuscript is now with our production department. 

With kind regards,

on behalf of

Dr. Guido Maiello 

Academic Editor

PLOS ONE